# The Impact of Quality of Digital Elevation Models on the Result of Landslide Susceptibility Modeling Using the Method of Weights of Evidence

**Mirosław Kamiński**

Polish Geological Institute–National Research Institute, Rakowiecka 4, 02-519 Warszawa, Poland; miroslaw.kaminski@pgi.gov.pl

**Abstract:** The paper discusses the impact that the quality of the digital elevation model (DEM) has on the final result of landslide susceptibility modeling (LSM). The landslide map was developed on the basis of the analysis of archival geological maps and the Light Detection and Ranging (LiDAR) digital elevation model. In addition, complementary field studies were conducted. In total, 92 landslides were inventoried and their degree of activity was assessed. An inventory of the landslides was prepared using a 1-m-LiDAR DEM and field research. Two digital photogrammetric elevation models with an elevation pixel resolution of 20 m were used for landslide susceptibility modeling. The first digital elevation model was obtained from a LiDAR point cloud (DEM–airborne laser scanning (ALS)), while the second model was developed based on archival digital stereo-pair aerial images (DEM–Land Parcel Identification System (LPIS)). Both models were subjected to filtration using a Gaussian low-pass filter to reduce errors in their elevation relief. Then, using ArcGIS software, a differential model was generated to illustrate the differences in morphology between the models. The maximum differences in topographic elevations between the DEM–ALS and DEM–LPIS models were calculated. The Weights-of-Evidence model is a geostatistical method used for the landslide susceptibility modeling. Six passive factors were employed in the process of susceptibility generation: elevation, slope gradient, exposure, topographic roughness index (TRI), distance from tectonic lines, and distance from streams. As a result, two landslide susceptibility maps (LSM) were obtained. The accuracy of the landslide susceptibility models was assessed based on the Receiver Operating Characteristic (ROC) curve index. The area under curve (AUC) values obtained from the ROC curve indicate that the accuracy of classification for the LSM–DEM–ALS model was 78%, and for the LSM–LPIS–DEM model was 73%.

**Keywords:** landslides; landslide susceptibility; geostatistics; weights-of-evidence; digital elevation model

---

## 1. Introduction

Landslides are a significant geodynamic threat in many areas of the world. They cause major economic losses as well as threaten human life [1]. The Flysch Carpathians area in Poland, especially the Dynów Foothills, is predisposed to the formation of various types of mass movements, mainly landslides [2–4]. The purpose of this research was to develop an optimal model of landslide susceptibility, taking into account digital photogrammetric elevation models of different qualities, which were used for the modeling process. For this purpose, two digital elevation models were used. The first was developed based on filtration and classification of the cloud of elevation points originating from aerial laser scanning (ALS), which is hereafter referred to as DEM–ALS [5]. The other model was developed based on digital stereo-pairs of aerial images, which is hereafter referred to as DEM–LPIS (Land Parcel Identification System) [6–10]. The general availability of DEM–ALS and airborne imagery derived from DEM in

geodetic centers in Poland was the reason for using these for the comparison process. A 20 m resolution was chosen for this study because global research into the impact of the resolution on the landslide susceptibility model shows this to be optimal [11]. In addition, the quality of the airborne imagery was low, which made it difficult to obtain greater detail.

When looking through the literature, it can be seen that many researchers have focused on the impact of the digital elevation model resolution on the final correctness of landslide susceptibility classifications [12,13]. It was found that DEM resolution is an important criterion and that it can significantly affect the accuracy of landslide susceptibility maps. It was noticed that the higher the DEM spatial resolution, the more strongly the results depended on topographic details [14–16]. It is worth noting that the digital elevation model (DEM) is also a key part of the basic spatial data used to analyze landslide susceptibility. The digital elevation model is the basis for generating factors determining the formation of landslides. These include, among others, the slope angle, the spatial orientation of slopes, the elevation, the curvature, and the topographic roughness index (TRI) [17]. DEMs are often generated using data obtained from different remote sensors, including optical imaging sensors, light detection and ranging (LiDAR), and synthetic aperture radar (SAR) [18]. The qualities of DEM-derived factors often depend on the spatial resolution of DEMs. This has been widely discussed in the literature [19]. Therefore, the choice of DEM is important for the assessment of landslide susceptibility.

The quality of the DEMs is essential for assessing their suitability and determines the quality of the geomorphometric analysis [20–22]. As a result of the variety of available DEMs, it is necessary to investigate their use in different scenarios. It is necessary to remember that working with digital data requires paying particular attention to their quality. Small errors in DEMs can produce large errors in the derived elevation attributes [23], especially second-order derivatives such as curvature [24]. DEM accuracy depends on the type of topography and ruggedness of the elevation, as well as the type of vegetation [25], the methods for collecting elevation data, the method for DEM generation, the type of DEM grid, and DEM resolution [26,27]. The issue of error analysis in DEMs is relevant and discussed in the literature [28,29]. This study focuses on the usefulness of different DEMs with the same resolution for landslide susceptibility studies. These analyses consisted of checking their vertical accuracy.

The impact that the quality of digital elevation models based on various sources of data has on landslide susceptibility models has not yet been widely discussed in the professional literature. In my opinion, landslide susceptibility modeling is more influenced by the quality of data (especially in digital elevation models) than the number of factors used.

The Weights-of-Evidence (WoE) method is a geostatistical method used for landslide susceptibility modeling [30–38]. This is a logarithmic form of the Bayesian probability model, often used in geomorphological risk assessment [33,39–43]. Two-dimensional statistical analysis methods, such as WoE, are amongst many of those used to assess landslide susceptibility. GIS-based geostatistical methods have become very popular in evaluating landslide susceptibility [44–56]. The Receiver Operating Characteristic (ROC) curve was used to assess the quality of landslide susceptibility classification maps [57,58]. Therefore, this research hopes to contribute to the scientific discussion on the problem presented in the article.

## 2. Study Area

The study area, the Dynów Foothills, is located in southeastern Poland in the San River valley (Figure 1). The topography of the area follows its geological structure. The elevation relief features wide ridges rising to elevations exceeding 400 m above sea level. The slopes of the ridges are cut by stream valleys. The upstream areas are usually trough-like and pass downstream into V-shaped gully valleys. Their lower reaches have narrow and flat bottoms. The slopes of the ridges are usually convex or straight and gently inclined at an angle of up to 20°, except in the headwaters areas, where they are trough-shaped. The relief is enriched with numerous landslides. In terms of geological structure, the Dynow Foothills are built of overthrust and folded flysch formations of the Skole Nappe [59].

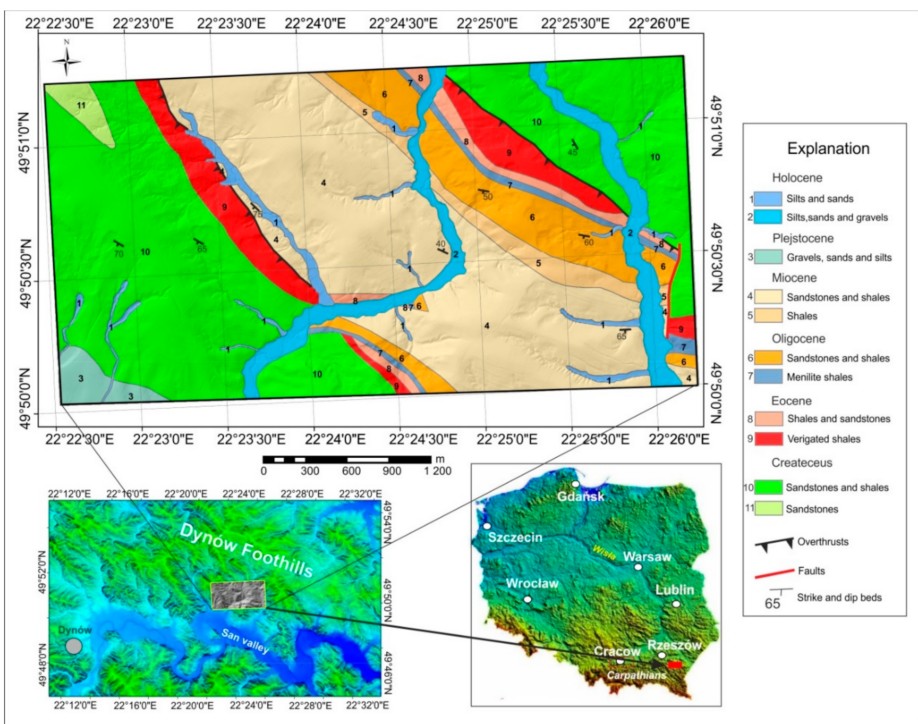

**Figure 1.** Study area location and geological setting.

Thick covers of Quaternary deposits, which favor the formation of shallow landslides, occur in the study area. These are mainly tills of various origins, as well as loesses. Their thicknesses are variable and can be up to 10 m. In addition, the flysch rock lithology accompanied by tectonic features promotes the formation of structural landslides. In the discussed area, the youngest flysch rocks are Oligocene shales and sandstones of the Krosno Beds. They consist of carbonate-cemented sandstones (Figure 1). They are interbedded with grey marly shales. These layers are underlain by Eocene variegated shales represented by clayey red, green, grey and spotted rocks, i.e., the so-called variegated shales [59]. They are predisposed to the formation of landslides due to their physical properties (cleavage, plasticity, impermeability). The variegated shales are underlain by the Cretaceous sandstone–shale flysch. The sandstones are usually thin-bedded and interlayered with ashy grey mudshales [59]. As a result of its physical properties, the flysch is highly susceptible to landsliding. In terms of tectonic structure, the research area is located within the overthrust zone of the Jawornik–Dubiecko overturned fold onto the Chyrzynka–Huta syncline. This zone makes this region of the Dynow Foothills more susceptible to the formation of landslides by weakening the structure of the rocks and allowing rainwater migration within them.

## 3. Methods and Materials

### 3.1. Archival Digital Stereo-Pair Aerial Images

In order to precisely determine the extent of a landslide and to present its surface relief, an accurate photogrammetric digital elevation model was used. It was purchased at the National Surveying and Cartographic Documentation Centre. Ten digital stereo-pairs of black-and-white aerial images, at a scale of 1:10,000, taken in 2009 as part of the Land Parcel Identification System (LPIS) project, were used to develop the model. The digital elevation model was generated on the Dephos photogrammetric station using an automatically generated 20 × 20 m grid of elevation points. The grid was subjected to stereoscopic editing consisting in determining the values of the grid points (the so-called pickets). The measured values of the height of the grid pickets were enriched with structural relief lines, such as scarp edges, lines of ridges and watercourses, and areas excluded due to dense vegetation. Using the

Inverse Distance Weight (IDW) algorithm, a digital elevation model was generated in a 20 × 20 m grid structure. It is very common to use an algorithm to interpolate the spatial data [60]. The average fitting error for the photogrammetric model in the grid is presented in Table 1.

**Table 1.** Summary of the results of aerial triangulation.

| RMS x (m) | RMS y (m) | RMS Total (m) | The Number of Aerial Images | Year |
|-----------|-----------|---------------|------------------------------|------|
| 0.44 | 0.36 | 0.32 | 12 | 2009 |

### 3.2. Airborne Laser Scanning (ALS)—LiDAR Data

The elevation relief was examined based on elevation data from the IT system of the country's protection against extraordinary threats (ISOK, http://www.gugik.gov.pl/projekty/isok). The products that LiDAR (Light Detection and Ranging) obtained from that project were made available by the National Surveying and Cartographic Documentation Centre in Warsaw. The data acquired during laser scanning are saved as files in the ASPRS Accuracy Standards for Digital Geospatial Data's Laser File Format (LAS). The scanning density was four points per square meter. The basic procedures for processing the point cloud obtained as a result of airborne laser scanning (ALS) are their classification and filtration [61]. The elevation data were acquired in September 2014. The process of generating DEM was based on the point cloud classification. The point cloud was divided into the following layers: ground, low vegetation, medium vegetation, high vegetation, and buildings. Then, the points representing the ground were filtered from the point cloud, using the tools available in the QCoherent software. The next step was the interpolation of elevation points using the algorithms. Two algorithms were tested: Inverse Distance Weighting (IDW) and the Adaptive Triangulated Irregular Network (TIN) model algorithm. Both algorithms provide a similar classification for point clouds describing land use for agriculture, e.g., areas on which a single building, shrub, or tree is located. The Adaptive TIN algorithm works better than the IDW algorithm in terms of the points recorded by the laser beam being reflected from ground, vehicles, and bridges. The Adaptive TIN Ground Point Cloud Task is an automated algorithm designed to separate points that have a high probability of being ground points from other points. The algorithm divides the task area into cells whose X and Y dimensions are defined by a seed parameter. For each cell, a "best" candidate ground point is selected. These "seed" points are then used to construct a Triangulated Irregular Network (TIN). The algorithm then iterates, attempting to add additional points to the TIN, based on several inclusion criteria. The iterations conclude when either the user-specified stopping criteria is met or no additional points were added during the previous iteration. As a result, two digital elevation models were developed in the grid structure at mesh sizes of 1 mm and 20 m. The first was used to identify landslides, the second for landslide susceptibility modeling.

### 3.3. Gaussian Low-Pass Filter

On the basis of the literature [62], the Gaussian low-pass filter was selected for further generalization of the surface of both digital elevation models. Low-pass filters are the averaging filters with certain weights. By adjusting the filter matrix row correctly, you can remove disturbances and errors of varying magnitudes from a digital elevation model of grid structure. To generalize the surface of the digital elevation models, four Gaussian low-pass filters were tested: STd_dev_0.391.ker; STd_dev_0.625.ker; STd_dev_1.0.ker; and STd_dev_1.6.ker, which are available in the ER Mapper software. The best results of the digital elevation model generalization were obtained after applying the STd_dev_1.6.ker 9 filter [63]. The obtained results of elevation generalization were compared in the ArcGis software with the hypsometry of the topographic map. It was assumed that the topographic map at the scale of 1:10,000 is a sufficiently accurate reflection of the topography+. The operation of the Gaussian low-pass filter relies on removing details from the image (e.g., differences between adjacent pixels), thereby smoothing the image. This filter was used to generalize the relief in both the DEM–ALS and DEM–LPIS digital elevation models.

### 3.4. Weights-of-Evidence Method

The WoE method is one of the multi-criteria decision-making methods. It is a log-linear version of Bayesian General Theory. This method has been used in many fields of science, e.g., in medical diagnostics and geology (determining potential sites of mineral deposits) [64–67], as well as for determining slopes susceptible to landsliding [68]. In probability theory and statistics, Bayes' theorem describes the probability of an event, based on prior knowledge of conditions that might be related to the event.

This is expressed by the formula

$$P(L/F) = \frac{P(L \cap F)}{P(F)} = \frac{P(L) \cdot P(F/L)}{P(F)} \tag{1}$$

where

$P(L)$ is an a priori probability of a given phenomenon (landslide occurrence probability);

$F$ is a factor determining the occurrence of a given phenomenon;

$P(F/L)$ is a conditional probability;

$P(L/F)$ is a probability that a given phenomenon will occur due to factor $F$ (a posteriori).

The a priori probability can be defined as the ratio of the area occupied by the landslide to the entire study area.

For practical reasons, the odds expression is introduced, which is defined as the ratio of the probability of occurrence of a given phenomenon to the probability of its not occurring.

This is expressed by the formula

$$Odds = O = \frac{P(L)}{1 - P(L)}. \tag{2}$$

As a result of further transformations, conditional odds are determined, and then the so-called association indicators (weights of "premises") $W+$ and $W-$, i.e., Weights-of-Evidence (which gave the name to the method), are calculated based on their logarithmic quantities. The difference between $W+$ and $W-$ is referred to as contrast (C). This is one of the elements, which determines the acceptance or rejection of a class of variables adopted for modeling.

$$W+ = \ln \frac{P\{B/D\}}{P\{B/D\}} \tag{3}$$

$$W- = \ln \frac{P\{B/D\}}{P\{B/D\}} \tag{4}$$

where

$P$ is a landslide occurrence probability;

$B$ is a factor determining the occurrence of a given phenomenon;

$D$ is a factor indicating the existence or absence of landslide.

The $W+$ positive weight shows more evidence than chance alone and is, therefore, directly proportional to the influence that the respective variable has on the likelihood of landslide occurrence. Contrast (C) is the difference between two weights (C = $W+ - W-$). This is one of the elements that determines the acceptance or rejection of a class of variables adopted for modeling.

## 4. Results

### 4.1. Landslide Inventory

The landslide inventory was made using archival digital aerial images and a DEM–ALS [69]. Supplementary field studies were also carried out [70,71]. Ninety-two landslides with a total area of 225,182 m$^2$ were surveyed and their activity model developed (Figure 2). In order to develop a model

of landslide activity, three field trips were made. Active slopes of landslides were located using a Real Time Kinematic (RTK GPS) receiver. The new fractures observed both on the landslide surface and on buildings and roads were the main criteria for landslide activity. Many landslides had new fractures in the main and minor scarps. This mainly indicated the activity of landslides.

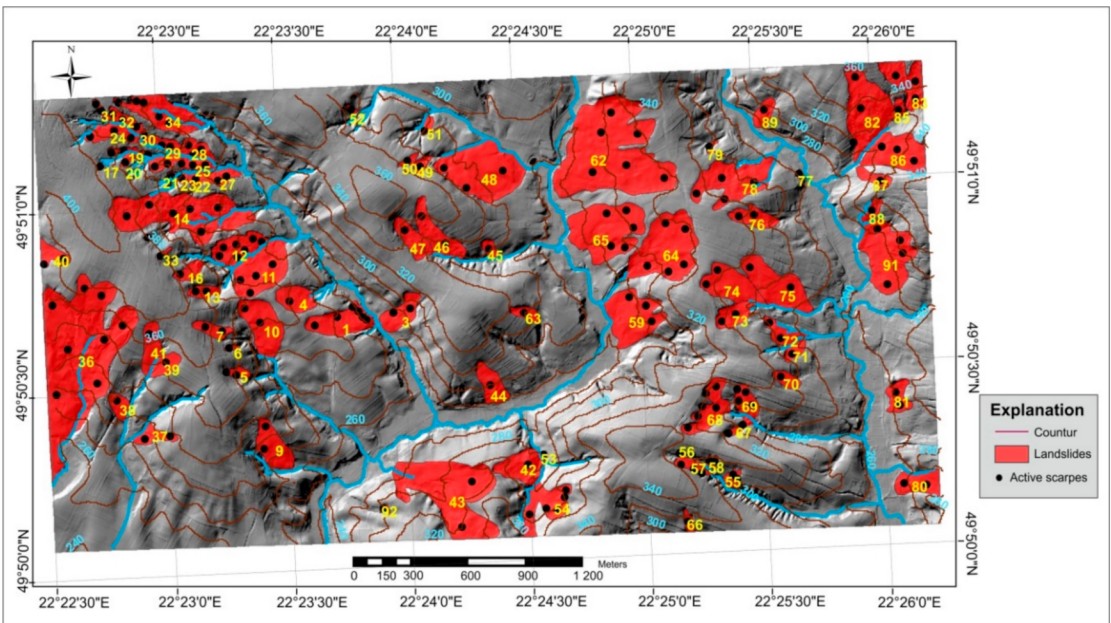

**Figure 2.** Landslide distribution map and active scarp.

For comparison purposes, the landslide ratio/index and the landslide density index were calculated [2].

$$Op = \frac{Po}{Pt} \times 100\%$$

where

    $O_p$—Landslide ratio/index [%];
    $P_O$—Landslide area [km$^2$];
    $P_t$—Surface area [km$^2$].

$$G = \frac{n}{Pt}$$

where

    $G$—Landslide density ratio/index [number/km$^2$];
    $n$—Number of landslides [number];
    $P_t$—Surface area [km$^2$].

The landslide ratio/index was 24.7%, while the landslide density ratio/index was 1.316/km$^2$ [2]. For comparison, the density ratio/index given in the literature for the Dynow Foothills area was 0.193/km$^2$ [2]. The reason for such a large difference was the greater number of inventory landslides.

The largest landslide clusters occur in the western and northeastern parts of the study area. In the western part, the largest landslide, number 36, registered in the western part of the study area, covers an area of over 2.5 ha (The actual size of this landslide is much greater because only its northeastern part extends into the area under study. It is developed totally within the Inoceramian Beds of Cretaceous flysch.

Most of the landslides occur in the northwestern part of the study area. These are obsequent landslides developed mainly within a dense network of stream valleys overgrown with dense forests (Figure 3). Many of these landslides threaten local roads and buildings (Figure 4). There may be a phenomenon of stream capture between landslides number 10 and number 11 because their main

slopes are only about 3 m apart. Landslides numbers 1 and 3 slide downslope to a stream whose flow direction in the channel is strongly disturbed by the advancing landslide toes. As a result of their increased activity, the stream might be blocked by the toes of both landslides and a dam lake might form. In relation to the strata dip, landslide number 3 is a consequent landslide. Its length is over 240 m and it covers an area of 2.7 ha (Figure 5). The main scarp is about 20 m high. It developed within flysch deposits of the Krosno Beds. The colluvial material consists of tills with rock debris. It threatens two residential buildings located near the landslide. Over the last 10 years, the landslide has not shown significant activity. Landslide number 1 is located on the southwestern slope. It is a typical rotational landslide, i.e., the sliding rock masses have rotated. As a result of the movement direction of rock material in relation to the position of bedrock layers, it is an obsequent landslide. Its length is over 350 m and its area is about 3.5 ha. The main scarp exceeds 1.5 m in height. The landslide developed between the Eocene variegated shales and the Oligocene flysch of the Krosno Beds. It was activated in spring 2001. It activated as a result of the springtime melting of wet snow on the slope for a long period. As a result, it damaged a local road and destroyed one outbuilding. A number of other residential buildings found themselves in a zone of immediate high risk of damage. Most of the landslides in this part of the study area are not large in terms of surface area. However, they are mostly active and show clearly developed main scarps and numerous, recently formed fractures.

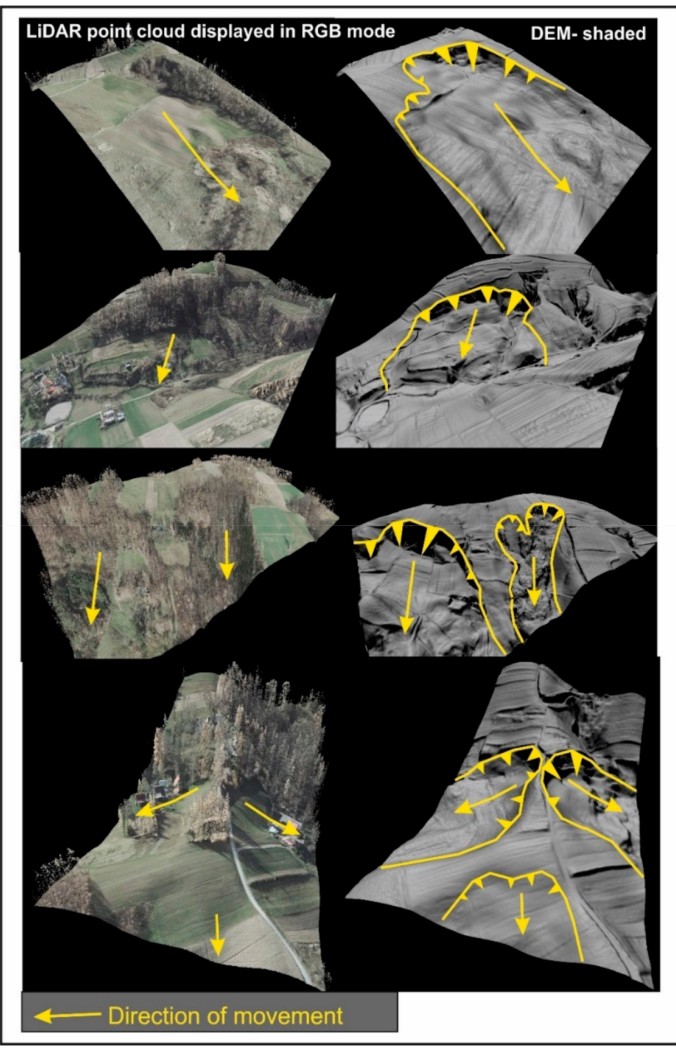

**Figure 3.** Landslides in a Light Detection and Ranging (LiDAR) image before and after filtration.

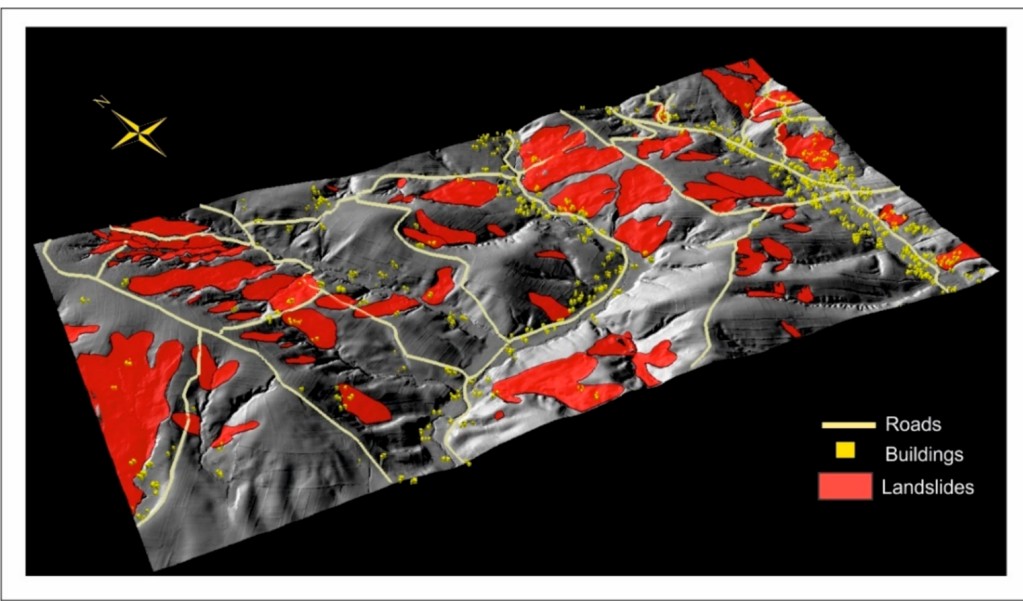

**Figure 4.** The locations of buildings and roads against the background of landslide settlement.

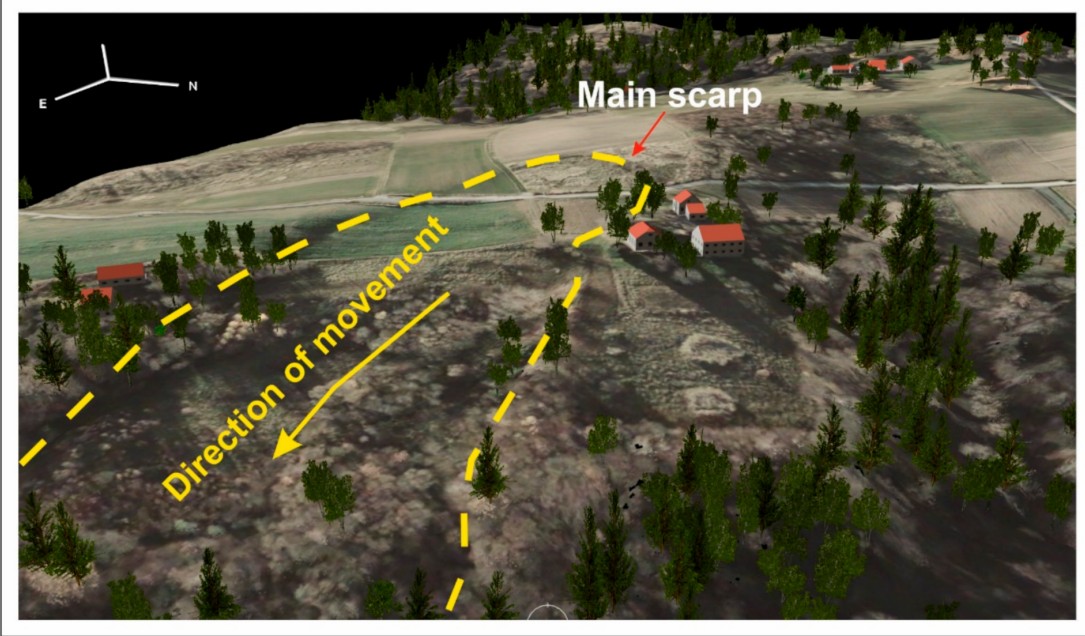

**Figure 5.** 3D visualization of landslide number 1. Buildings that are endangered by the landslide are visible. Visualizations were developed in ENVI LiDAR software using point cloud filtration products.

For these reasons, they pose a great threat to local infrastructure. Landslides located in the northeastern part of the research area are larger than those found in its western part. The largest of them is landslide number 62, which occupies an area of over 2 ha. It threatens local roads and numerous residential and utility buildings. It is a subsequent landslide that formed on a boundary between four lithological complexes: Cretaceous shale–sandstone flysch of the Inoceramian Beds, Eocene variegated shales and shale flysch of the Hieroglyphic Beds, and shale–sandstone flysch of the Oligocene Krosno Beds. The other landslides are slightly smaller; in most cases, they are classified as obsequent and subsequent landslides with distinct main scarps. Some of these landslides were formed in forests, e.g., number 86 and number 82, while others developed within arable fields and meadows, which is why they are a big problem for agriculture and forestry.

### 4.2. Landslide Conditioning Factors and Their Analysis

Landslide susceptibility modeling comprised four main stages. The first stage consisted of collecting data and constructing the GIS database. These were the layers with landslides and nine predictive factors: lithology, faults, streams, elevation, slope gradient, spatial orientation of slopes, curvature, roughness index (TRI), and terrain position index (TPI) (Figure 6). On the basis of field research and an analysis of archival materials, a model of landslide activity was developed. One hundred and ninety-three active landslide scarps were identified. Then, the dataset was divided randomly into two subsets. The first subset of data (97 slopes) was used to develop susceptibility models, while the second subset of data (96) was used to verify these models.

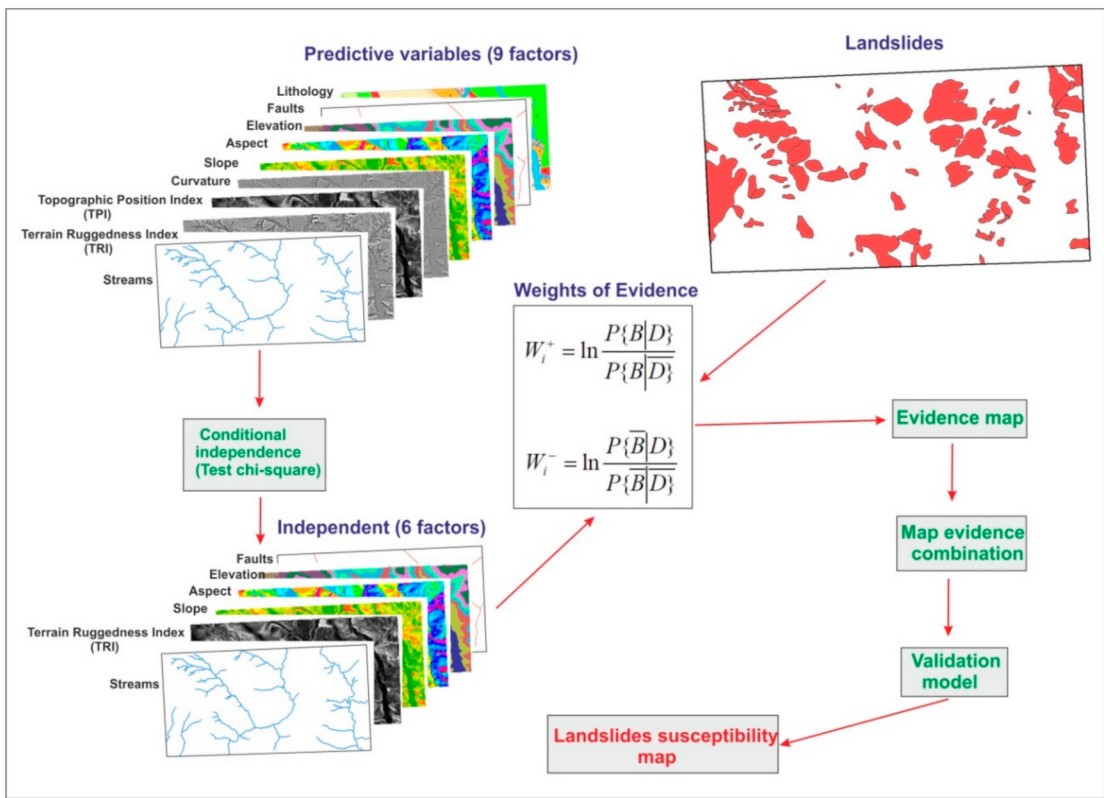

**Figure 6.** Diagram of a landslide susceptibility map development.

The modeling was carried out using the ArcSDM (Spatial Data Modeler) module, which is an extension of ArcGIS, developed by the Geological Survey of Canada [72]. This module automatically calculates test chi-square positive and negative weight values, variances, contrast, and posteriori probability (Tables 2 and 3). The first three factors were created in a vector form in the geobase, while the remaining six were developed as a result of DEM–ALS and DEM–LPIS processing. All layers were converted to a raster format in a grid structure, with a 20 m pixel size. In the second stage, a chi-squared test of independence was performed for the nine predictive factors. The test chi-square is based on a comparison of the observed values (obtained in the study) and theoretical values (calculated assuming that there is no relationship between the variables). Large differences indicate the existence of dependencies between the variables [32].

**Table 2.** Spatial relationship between each landslide conditioning factor and landslide using the LSM–ALS model.

| Class | Area (km$^2$) | Points | W+ | s (W+) | W− | s (W−) | C | s (C) | Stud (C) |
|---|---|---|---|---|---|---|---|---|---|
| **Distance to fault (m)** | | | | | | | | | |
| 0–50 | 6266 | 2 | −0.2108 | 0.7098 | 0.0049 | 0.1025 | −0.2157 | 0.7171 | −0.3008 |
| 50–100 | 6760 | 1 | −1.1767 | 1.0014 | 0.0234 | 0.1020 | −1.2001 | 1.0066 | −1.1922 |
| 100–150 | 7187 | 2 | 0.2091 | 0.7111 | −0.0039 | 0.1025 | 0.2130 | 0.7185 | 0.2964 |
| 150–200 | 7418 | 0 | | | | | | | |
| >200 | 7578 | 3 | 0.6111 | 0.5823 | −0.0143 | 0.1031 | 0.6254 | 0.5913 | 1.0577 |
| **Elevation (m.a.l.s)** | | | | | | | | | |
| 230–235 | 1184 | 1 | −2.3963 | 1.0004 | 0.1086 | 0.1021 | −2.5048 | 1.0056 | −2.4909 |
| 235–255 | 2368 | 7 | −1.1414 | 0.3785 | 0.1790 | 0.1054 | −1.3204 | 0.3929 | −3.3605 |
| 255–312 | 3552 | 12 | −1.0074 | 0.2892 | 0.2778 | 0.1085 | −1.2853 | 0.3088 | −4.1615 |
| 312–363 | 4736 | 28 | −0.4453 | 0.1895 | 0.2541 | 0.1202 | −0.6993 | 0.2245 | −3.1155 |
| 363–415 | 5920 | 46 | −0.1701 | 0.1480 | 0.1792 | 0.1394 | −0.3492 | 0.2034 | −1.7176 |
| **Aspect** | | | | | | | | | |
| N | 1154 | 14 | 0.2537 | 0.2689 | −0.0367 | 0.1096 | 0.2904 | 0.2904 | 1.0001 |
| N–E | 2308 | 28 | 0.2539 | 0.1901 | −0.0859 | 0.1200 | 0.3398 | 0.2249 | 1.5111 |
| S–E | 3461 | 42 | 0.2540 | 0.1552 | −0.1555 | 0.1342 | 0.4094 | 0.2052 | 1.9953 |
| S–W | 4615 | 50 | 0.1393 | 0.1422 | −0.1271 | 0.1449 | 0.2664 | 0.2030 | 1.3121 |
| N–W | 5768 | 65 | 0.1790 | 0.1247 | −0.2798 | 0.1747 | 0.4588 | 0.2147 | 2.1374 |
| **Slope (degrees)** | | | | | | | | | |
| 0–2 | 822 | 0 | | | | | | | |
| 2–5 | 2126 | 6 | −1.2140 | 0.4088 | 0.1677 | 0.1048 | −1.3816 | 0.4221 | −3.2735 |
| 5–13 | 3768 | 18 | −0.6856 | 0.2363 | 0.2505 | 0.1125 | −0.9361 | 0.2617 | −3.5773 |
| 13–20 | 5492 | 46 | −0.1205 | 0.1481 | 0.1203 | 0.1394 | −0.2408 | 0.2034 | −1.1841 |
| 20–54 | 7107 | 68 | 0.0138 | 0.1219 | −0.0305 | 0.1834 | 0.0443 | 0.2202 | 0.2012 |
| **Topographic Ruggedness Index** | | | | | | | | | |
| (−160)–(−2) | 1152 | 3 | −1.2960 | 0.5781 | 0.0876 | 0.1031 | −1.3836 | 0.5872 | −2.3561 |
| (−2)–(−1) | 2305 | 5 | −1.4788 | 0.4477 | 0.2010 | 0.1043 | −1.6798 | 0.4597 | −3.6543 |
| (−1)–0 | 3457 | 14 | −0.8527 | 0.2678 | 0.2540 | 0.1098 | −1.1068 | 0.2894 | −3.8240 |
| 0–1 | 4610 | 28 | −0.4452 | 0.1896 | 0.2540 | 0.1203 | −0.6993 | 0.2245 | −3.1150 |
| 1–12 | 5762 | 50 | −0.0859 | 0.1420 | 0.0981 | 0.1451 | −0.1841 | 0.2030 | −0.9066 |
| **Distance to streams (m)** | | | | | | | | | |
| 0–55 | 684 | 1 | −1.8469 | 1.0007 | 0.0566 | 0.1020 | −1.9036 | 1.0059 | −1.8924 |
| 55–69 | 912 | 10 | 0.1780 | 0.3180 | −0.0184 | 0.1071 | 0.1963 | 0.3355 | 0.5851 |
| 69–90 | 356 | 5 | 0.4293 | 0.4504 | −0.0186 | 0.1042 | 0.4479 | 0.4623 | 0.9690 |
| 90–120 | 491 | 7 | 0.4430 | 0.3807 | −0.0272 | 0.1053 | 0.4701 | 0.3950 | 1.1902 |

**Table 3.** Spatial relationship between each landslide conditioning factor and landslide using the LSM–LPIS model.

| Class | Area (km²) | Points | W+ | s (W+) | W− | s (W−) | C | s (C) | stud (C) |
|---|---|---|---|---|---|---|---|---|---|
| **Distance to fault (m)** | | | | | | | | | |
| 0–50 | 6266 | 61 | −0.0023 | 0.1287 | 0.0093 | 0.2595 | −0.0116 | 0.2896 | −0.0399 |
| 50–100 | 6760 | 66 | 0.0006 | 0.1237 | −0.0039 | 0.3178 | 0.0045 | 0.3410 | 0.0131 |
| 100–150 | 7187 | 70 | −0.0019 | 0.1201 | 0.0221 | 0.4103 | −0.0240 | 0.4275 | −0.0561 |
| 150–200 | 7418 | 72 | −0.0054 | 0.1184 | 0.1025 | 0.5027 | −0.1079 | 0.5165 | −0.2089 |
| >200 | 7578 | 74 | 0.0008 | 0.1168 | −0.0283 | 0.7105 | 0.0291 | 0.7200 | 0.0404 |
| **Elevation (m.a.l.s)** | | | | | | | | | |
| 230–235 | 903 | 1 | −2.1775 | 1.0006 | 0.0841 | 0.1021 | −2.2616 | 1.0057 | −2.2487 |
| 235–255 | 2267 | 9 | −0.8978 | 0.3340 | 0.1591 | 0.1066 | −1.0568 | 0.3506 | −3.0144 |
| 255–312 | 3786 | 18 | −0.7168 | 0.2363 | 0.2689 | 0.1125 | −0.9857 | 0.2617 | −3.7667 |
| 312–363 | 5333 | 41 | −0.2331 | 0.1568 | 0.2093 | 0.1332 | −0.4425 | 0.2058 | −2.1504 |
| 363–415 | 6845 | 63 | −0.0517 | 0.1266 | 0.1003 | 0.1699 | −0.1520 | 0.2119 | −0.7173 |
| **Aspect** | | | | | | | | | |
| N | 837 | 7 | −0.1670 | 0.3796 | 0.0142 | 0.1059 | −0.1812 | 0.3941 | −0.4599 |
| N–E | 2236 | 29 | 0.2766 | 0.1869 | −0.0981 | 0.1218 | 0.3747 | 0.2231 | 1.6794 |
| S–E | 3412 | 43 | 0.2474 | 0.1535 | −0.1609 | 0.1367 | 0.4083 | 0.2055 | 1.9869 |
| S–W | 4578 | 53 | 0.1614 | 0.1382 | −0.1651 | 0.1514 | 0.3265 | 0.2050 | 1.5931 |
| N–W | 6019 | 66 | 0.1066 | 0.1238 | −0.1950 | 0.1803 | 0.3016 | 0.2187 | 1.3787 |
| **Slope (degrees)** | | | | | | | | | |
| 0–2 | 807 | 0 | | | | | | | |
| 2–5 | 2062 | 7 | −1.0735 | 0.3786 | 0.1622 | 0.1060 | −1.2357 | 0.3932 | −3.1429 |
| 5–13 | 3681 | 18 | −0.7070 | 0.2363 | 0.2669 | 0.1132 | −0.9739 | 0.2620 | −3.7170 |
| 13–20 | 5311 | 44 | −0.1765 | 0.1514 | 0.1747 | 0.1382 | −0.3512 | 0.2050 | −1.7134 |
| 20–54 | 6804 | 70 | 0.0422 | 0.1201 | −0.1018 | 0.1933 | 0.1440 | 0.2276 | 0.6325 |
| **Topographic Ruggedness Index (TRI)** | | | | | | | | | |
| (−160)–(−2) | 782 | 2 | −1.2860 | 0.8780 | 0.0576 | 0.2031 | −1.6837 | 0.5630 | −2.1321 |
| (−2)–(−1) | 2078 | 10 | −0.7246 | 0.3170 | 0.1304 | 0.1078 | −0.8550 | 0.3348 | −2.5536 |
| (−1)–0 | 3734 | 20 | −0.6168 | 0.2242 | 0.2505 | 0.1147 | −0.8673 | 0.2518 | −3.4437 |
| 0–1 | 5332 | 44 | −0.1816 | 0.1514 | 0.1808 | 0.1382 | −0.3624 | 0.2050 | −1.7682 |
| 1–12 | 6778 | 67 | 0.0005 | 0.1228 | −0.0012 | 0.1835 | 0.0018 | 0.2208 | 0.0080 |
| **Distance to streams (m)** | | | | | | | | | |
| 0–55 | 684 | 1 | −1.8948 | 1.0007 | 0.0600 | 0.1021 | −1.9548 | 1.0059 | −1.9433 |
| 55–69 | 1582 | 8 | −0.6494 | 0.3545 | 0.0848 | 0.1060 | −0.7341 | 0.3700 | −1.9844 |
| 69–90 | 1931 | 12 | 0.4424 | 0.2896 | 0.0808 | 0.1084 | −0.5233 | 0.3092 | −1.6924 |
| 90–120 | 10168 | 98 | 0.0600 | 0.1021 | 0.8948 | 0.0007 | 1.9548 | 1.0059 | −1.9433 |

Layers such as lithology, curvature, and topographic position index (TPI) showed a conditional relationship. Therefore, in a later stage of modeling, these were eliminated. Ultimately, only six predictive factors were used: faults, streams, elevation, slope gradient, spatial orientation of slopes, and roughness index (TRI). The third stage covered the use of the Weights-of-Evidence method to weigh the individual predictors. The calculation of weights for predictive quantities includes crossing operations between raster maps and a point map of the occurrence of active landslide scarps. As a result, two maps of landslide susceptibility were generated. The fourth stage involved estimating the accuracy and validation of the obtained landslide susceptibility models.

### 4.3. Comparison of the Elevation Differences between the Digital Elevation Models (ALS and LPIS)

A comparison of the quality of the photogrammetric digital elevation models was made in ArcGis.10.6 software. Two digital elevation models, LPIS and ALS, with an elevation pixel resolution of 20 m, were used for the analysis. Both models were subjected to a low-pass Gaussian filtration process. Using the Arc Map tools, a differential elevation model between the DEM–ALS and DEM–LPIS models was generated (Figure 7). In the western part of the differential model, two areas (number 1 and number 2) were distinguished, where there are the largest discrepancies between the models. This research showed a spatial distribution of elevation changes between the DEM–ALS and DEM–LPIS models. Their maximum values ranged from −21 m to +15 m. Such considerable differences in elevations between the models are mainly caused by the masking effect of vegetation on the surface (Figure 8).

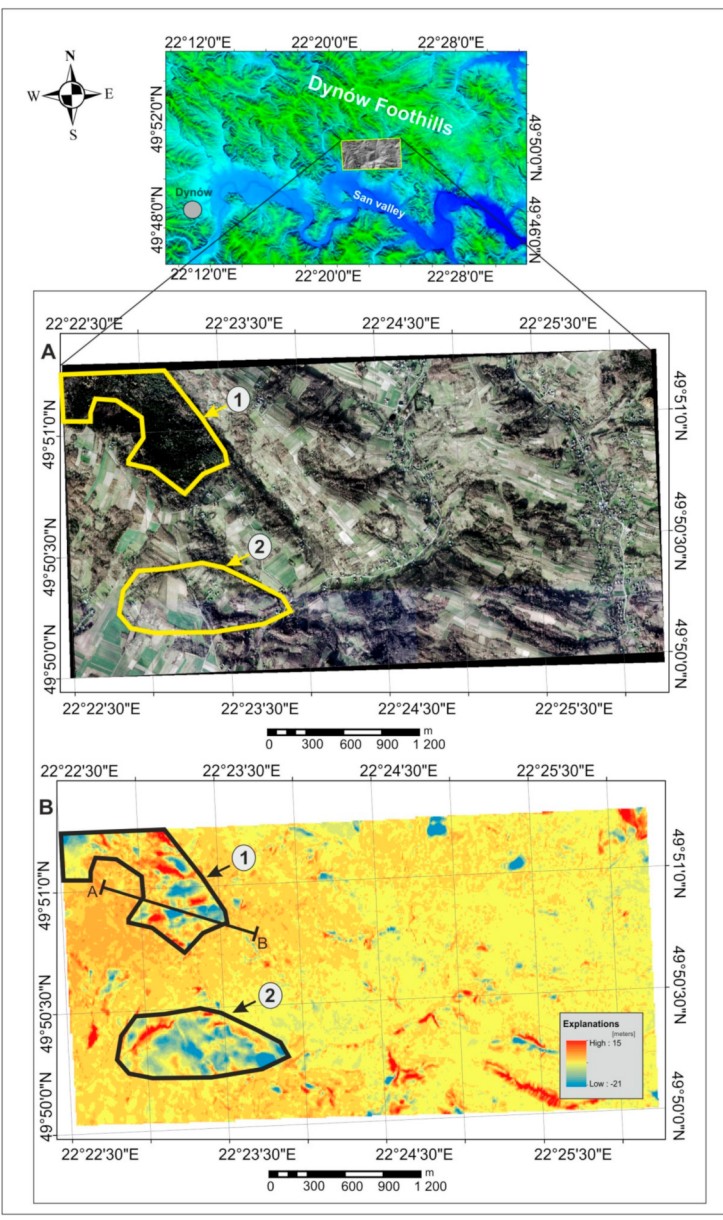

**Figure 7.** (**A**) Orthophotomap. (**B**) Differential elevation model comparing the DEM–ALS and DEM–LPIS models. The two areas highlighted are the areas with the largest differences in elevation between the DEM–ALS and DEM–LPIS models.

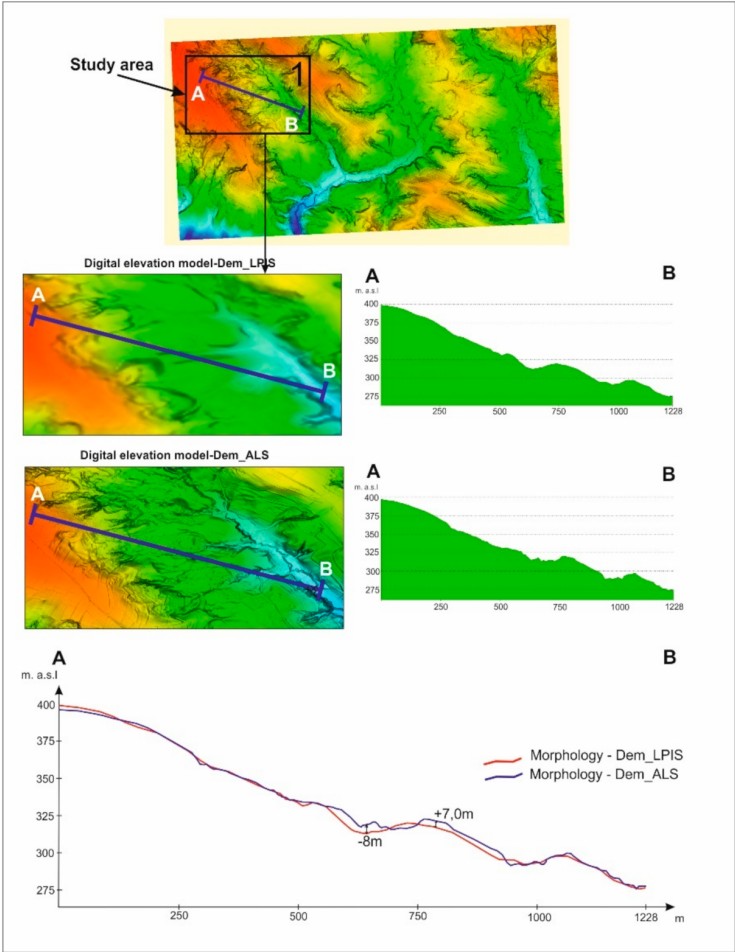

**Figure 8.** Morphological cross-section.

Area number 1 is marked on the orthophotomap. It is covered with dense vegetation. Area number 2 is more diverse in terms of vegetation cover. Meadows and arable land are distinguished here, in addition to high and medium vegetation. In other areas, covered mostly with forests, a significant increase in differences in elevations between both models was noted. In addition, a morphological cross-section was constructed for area number 1 (Figure 9). In the upper part of the cross-section, running through the meadows, slight differences in elevation of about 2 m were found. In turn, in the middle and lower parts of the cross-section, crossing the valleys of streams overgrown with trees, there was a significant increase in elevation differences. The maximum differences in the cross-section ranged from −8 m to + 7 m.

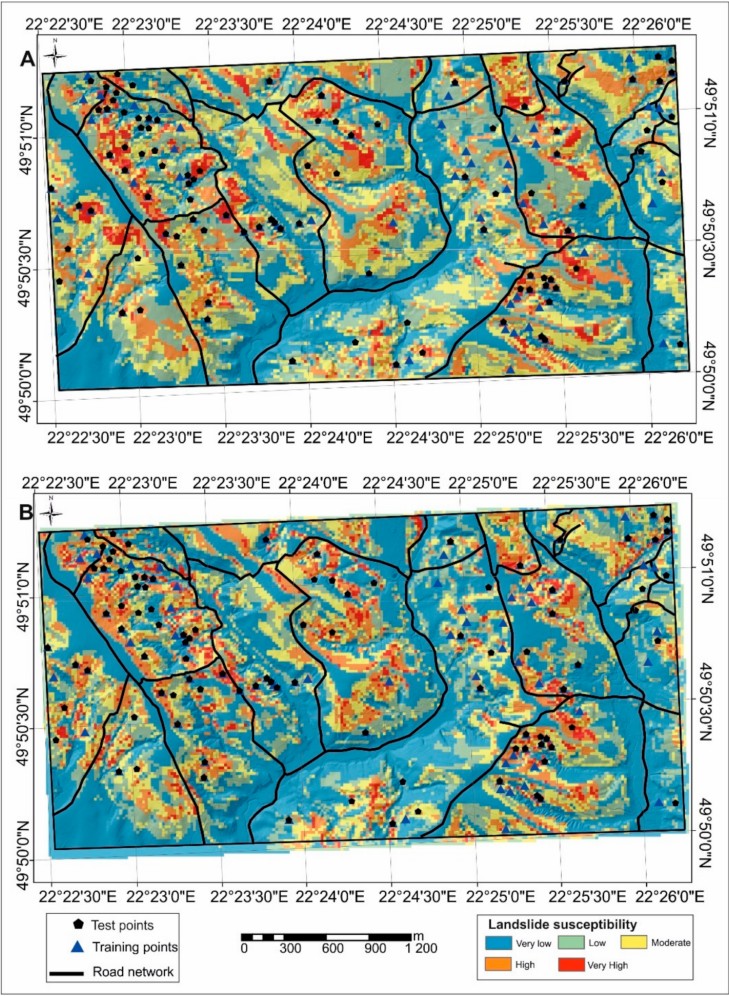

**Figure 9.** Landslide susceptibility maps against the road network. (**A**) LSM–ALS susceptibility, (**B**) LSM–LPIS susceptibility.

### 4.4. Landslide Susceptibility Maps

The susceptibility maps were developed by means of the Weights-of-Evidence geostatistical method using two different 20-m-resolution digital elevation models. The model allowed us to divide the study area into five morphodynamic categories: very weakly susceptible, weakly susceptible, moderately susceptible, highly susceptible, and very highly susceptible (Figure 9).

Stable areas, i.e., very poorly susceptible and poorly susceptible areas, are represented mainly by the stream valleys consisting of Holocene and Quaternary terraces. The same category refers to the flattening of hills, composed largely of Cretaceous sandstone–shale rocks and Oligocene shale–sandstone rocks. The remaining area was included in the category "unstable" (moderately, highly, and very highly susceptible). The most susceptible areas occur mainly on the hill slopes and predominantly follow the course of geological strata, mainly of the variegated shales and the Cretaceous shale–sandstone flysch of the Inoceramian Beds. There are several key differences between the LSM–ALS and LSM–LPIS landslide susceptibility models. These differences relate to the distribution and size of the areas occupied by individual susceptibility classes. The greatest number of the most weakly susceptible areas occurs in the LSM–LPIS model (Figure 10). Low and medium susceptibility areas occupy larger areas in the LSM–ALS model than in the LSM–LPIS model, whereas highly susceptible areas cover larger areas in the LSM–LPIS model than in the LSM–ALS model. Very highly susceptible areas occupy similar surface areas in both models (Figure 11).

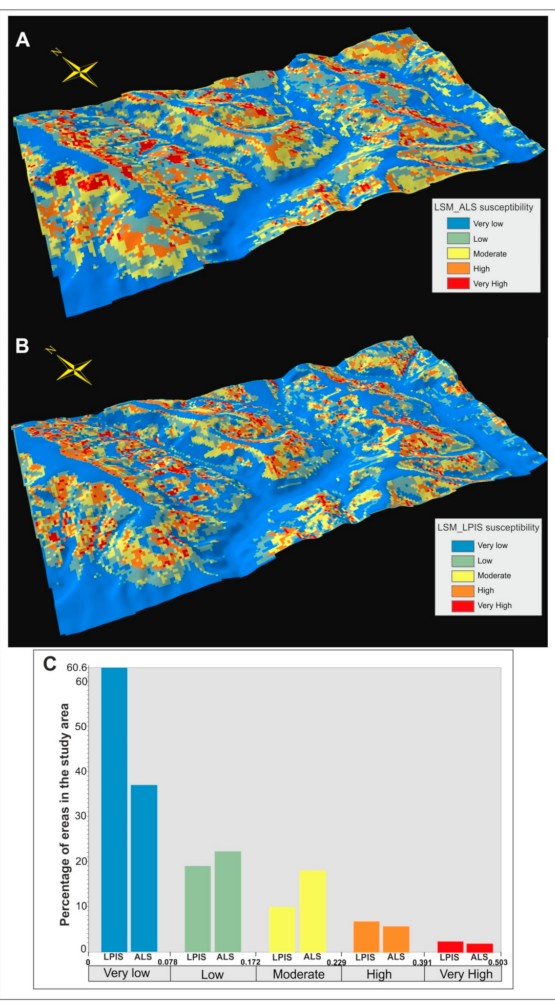

**Figure 10.** Spatial visualization of maps of landslide susceptibility (**A,B**). (**C**) Percentage list comparing landslide susceptibility for the LSM–ALS and LSM–LPIS models.

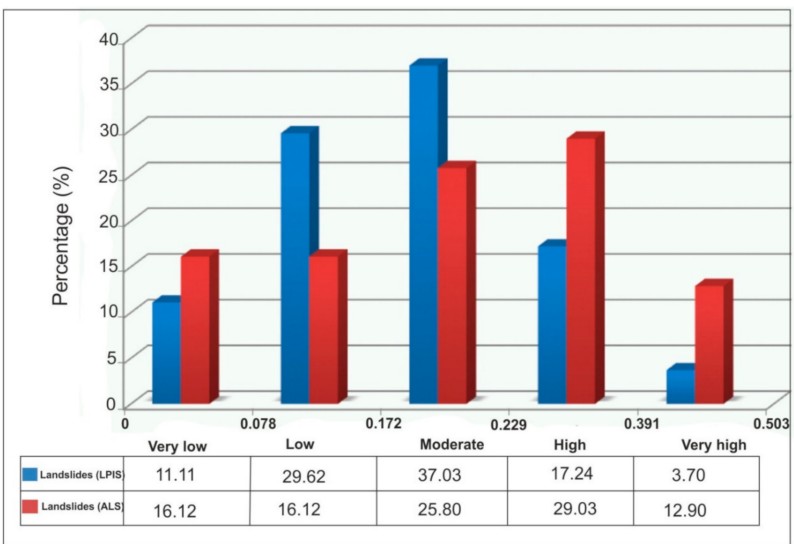

**Figure 11.** Histogram representing the distribution of observed landslides falling into various susceptibility classes of different landslide susceptibility zonation maps for the LSM–ALS and LSM–LPIS models.

The modeling results showed that the greatest number of landslides occurs in the classes "high" (12.9%) and "very high" (29.03%) in the LSM–ALS model (Figure 11).

## 5. Validation

The characteristics of the Receiver Operating Characteristic (ROC) curve were used to validate the two landslide susceptibility models obtained using the WoE geostatistical method. The ROC approach shows values between the true positive rate (TPR) and false positive rate (FPR) in determining the correctness of landsliding susceptibility. The y-axis indicates TPR, while the x-axis indicates FPR. The TPR is the pixels to which landslide areas were correctly assigned, while the FPR is the pixels to which landslide areas were incorrectly assigned. The Area Under Curve (AUC) is a measure indicating the accuracy of the landslide susceptibility models. The AUC values obtained from the ROC curve indicate that the classification accuracy was 78% for the LSM–DEM–ALS model and 73% for the LSM–LPIS–DEM model. The resulting ROC curve is shown in Figure 12.

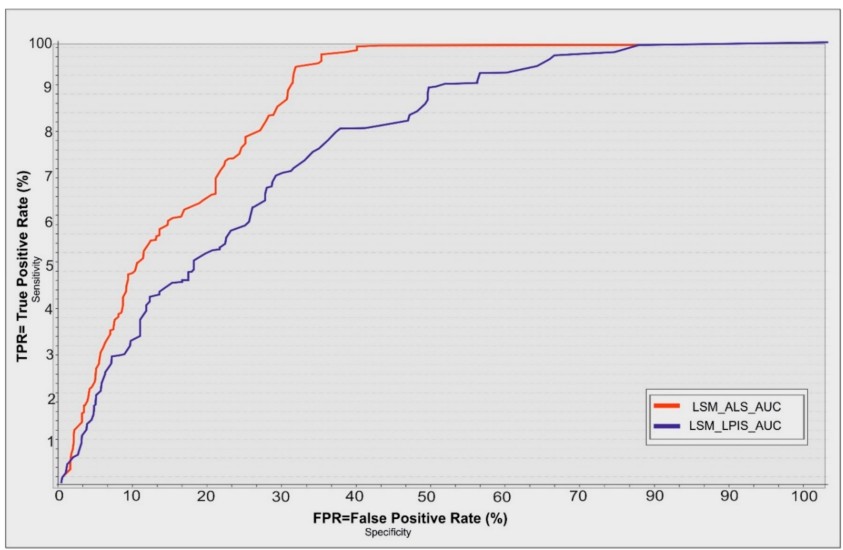

**Figure 12.** Validation of susceptibility maps.

## 6. Discussion

It is clear that the DEM–ALS and DEM–LPIS digital photogrammetric elevation models differ in terms of the accuracy of relief details. On the basis of the research, it should be stated that the DEM–ALS model is more accurate than the DEM–LPIS model. There are two main reasons for the quality deterioration of the DEM–LPIS model. The first is related to the methods of generating digital elevation models based on the digitization of stereo-pairs of aerial images. This process is carried out by the operator on a digital photogrammetric station. It is always a possibility that the operator will incorrectly measure both the elevation point values of the measuring grid and the structural lines of the relief, e.g., scarp edges and stream lines. The second reason for errors in the model is related to the dense vegetation covering the area, e.g., forests and shrubs, and compact settlements. The differential model comparing the DEM–ALS and DEM–LPIS models showed that the greatest differences occurred in areas covered with vegetation, which ranged from +15 m to −21 m and are considered large errors. The author is of the opinion that the elevation models generated digitally from aerial images are significantly burdened by the effect of land cover; therefore, they should be used to a limited extent in spatial analyses [73]. It should also be added that the DEM–ALS model is not without flaws. There are at least two factors that affect the accuracy of surface imaging. The first is the correct filtration of the points representing the elevation surface. In most cases, this is done automatically. However, in this way, points of low vegetation, for example, may be classified incorrectly as the surface [74]. The second factor is the selection of a method of elevation point interpolation. This is a very important stage in

generating digital elevation models, which requires a separate discussion. The present study used the Adaptive TIN algorithm, which is found in QCoherent software. The research presented in this paper refers to a wide research stream focused on the impact that the resolution and quality of digital elevation models has on the result of landslide susceptibility [75–78].

This study also demonstrated the usefulness of the Weights-of-Evidence geostatistical method in evaluating landslide susceptibility in mountainous areas [79,80]. The main advantage of this method is the ability to integrate many predictive factors and check, by means of the chi-square test, which are the most reliable for susceptibility modeling. The susceptibility maps differed in the distribution and size of areas occupied by individual landslide susceptibility classes. These differences were affected by the different digital elevation models used. The LSM–LPIS susceptibility model is characterized, for instance, by an uneven distribution of pixels representing high and very high susceptibility. This is an obvious error, which results from the poor quality of the DEM–LPIS digital model. Regarding the LSM–LPIS model, very poorly susceptible areas also covered over 60% of the study area. This percentage value seems to be too high as compared with 36% from the LSM–ALS model.

## 7. Conclusions

The Weights-of-Evidence method used in the study proved to be useful in landslide susceptibility modeling. Two digital elevation models, each with a 20 m pixel resolution, were employed for the modeling. The first model was developed on the basis of the classification, filtration, and interpolation of the elevation point cloud acquired from DEM–ALS aerial laser scanning, while the other was developed on a digital photogrammetric station based on digital stereo-pairs of aerial images at the scale of 1:10,000. Both models were filtered in the raster format with a Gaussian low-pass filter. The filter generalized the relief and reduced the impact of various artifacts on the accuracy of the models. A comparative analysis between the digital elevation models showed that the DEM–ALS model is more accurate than the DEM–LPIS model. The precision of photogrammetric measurements is mainly dependent on the quality of the aerial photographs. The photographic resolution is a function of the optical quality of an image and is influenced by the resolving power of the film and camera lens, image motion during exposure, atmospheric conditions, and the film processing conditions. The errors that occurred in the DEM–LPIS model were mainly due to the masking effect of vegetation, which effectively limits the ability to see details in the area examined. The second factor affecting the accuracy of the model was the subjective assessment of the elevations of the measuring grid points and the structural lines appearing in the aerial images, e.g., slope ridge lines and stream lines, by the operator of the photogrammetric station. It is worth noting that the density and type of vegetation may be a source of errors in both digital elevation models. On the other hand, errors in the DEM–ALS model may be generated both in the process of point cloud classification and its interpolation. Using the AUC curve, the classification accuracy was assessed to be 78% for the LSM–DEM–ALS model and 73% for the LSM–LPIS–DEM model. Roads, residential buildings, and forests fell within a range of very high- and high-risk classes. This research helps us to better understand the process by which various errors form and their impact on the final result of landslide susceptibility modeling. The presented methodology for developing landslide susceptibility maps has practical applications in terms of spatial planning for local self-government authorities and for banks, developers, and insurance companies. In order to improve the methodology, we plan to test other geostatistical and machine-learning methods in terms of their impact on the accuracy of landslide susceptibility evaluation. In addition, the effect of different resolutions of LiDAR data and methods of interpolation and filtration will be examined.

**Funding:** The final editorial work was supported by the PGI-NRI statutory funds (Project No. 62.9012.1942.00.0).

**Conflicts of Interest:** The author declares no conflict of interest. The funders had no role in the design of the study; in the collection, analyses, or interpretation of data; in the writing of the manuscript, or in the decision to publish the results.

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
