# Peer review of "The Impact of Quality of Digital Elevation Models on the Result of Landslide Susceptibility Modeling Using the Method of Weights of Evidence"

_geosciences, doi:10.3390/geosciences10120488_

Round 1
Reviewer 1 Report
Dear Author,
Thank you very much for your manuscript. You work is based on a comparison of two differently generated DEM`s (DEM-ALS and DEM-LPIS). The biggest advantage of ALS-DEM`s was already described at least 20 years ago ( e.g. SCHULZ, 2004). It is the ability of the LIDAR methodology to “see through” forested area and to create a digital map of the actual ground surface. It is a different methodical approach to create a DEM from stereo-pair aerial images, because in these images, the actual ground surface (especially in forested areas) can`t be seen. Of course, filters can be used, but in this case the actual ground surface is still only a calculated estimation. So, this is a methodical drawback of aerial photography, well known since at least the 1940`s.
I therefore don`t see any added value to the scientific community to compare the performance of a down-scaled (from 1m resolution to 20m !) DEM-ALS with a DEM-LPIS, especially in a heavily forested study area. Of course, DEM-LPIS will perform worse than DEM-ALS, and again - especially it will do so in the forested areas. That is why my overall recommendation to the editor is to reject your manuscript in this current form.
If you decide to rework your manuscript, I have to personal suggestions for a different thematical focus: One the one hand, your inventory data seems very promising. The density of your mapped landslides (No. and “total affected area” in relation to the size of your study area) is remarkable and you also seem to have enough background information to at least some of the landslides. On the other hand, your WofE modelling is very sound and performed state of the art. I was only wondering, why you just focused on only 9 predictor factors at the beginning (Line 227). There is a huge, ongoing scientific discussion, which predictor factors are useful – many related ideas can be found in recent publications (e.g. “distance to forests” could be very promising in your study area).
SCHULZ, W. H. (2004): Landslides mapped using LIDAR imagery, Seattle, Washington. US Geological Survey Open-File Report 2004-1396, 11 p., U.S. Department of the Interior, U.S. Geological Survey, Washington D.C..
Author Response
Dear Reviewer 1
Thank you for your honest review of the manuscript. Of course, you are right that a long time ago geomorphological research was conducted using LiDAR data and aerial photography. However, since then, new interpolating algorithms and better and better software have appeared. The methodology for generating digital elevation models from aerial images has also changed dramatically. The quality of aerial photos is getting better and the software for their processing includes many new tools. I tried to use all the latest available tools and algorithms in my work. In my case, it was mainly about tracing the influence of various errors on the final result of the estimation of landslide susceptibility. A fatal error has occurred. I did not use LiDAR Dem with 1m resolution for comparison, only 20m resolution. The digital terrain model generated from aerial images does not necessarily have to be worse for modeling landslide susceptibility than the LiDAR model. It depends on the quality of aerial photos and the methodology of their further processing. As for your suggestion to use other modeling factors. This is a good idea. However, it is worth remembering that in the WoE method, the chi square test determines the number of factors used. For this reason, for example, I had to give up the lithology factor. In my opinion, rock lithology has a greater influence on landslide susceptibility than the distance from the forest. I intend to test several machine learning methods using multiple factors in my next research projects.

Reviewer 2 Report
- Line 176, superscript of m2
- Table 1 and Table 2 is not easy to read. Lines between columns will be helpful.
- The original data of DSM(DEM) is denser than 20m. Why use 20m resolution data for comparison?
- The spelling should be “ArcGIS”
- The derivation of calculation of landslide susceptibility is not clear and also the classification of susceptibility is not defined.
Author Response
Dear Reviewer 2
Thank you for the helpful review and relevant comments. I took into account all your comments. Especially point 5 I corrected. By answering question number two. The original data of DSM (DEM) is denser than 20m. Why use 20m resolution data for comparison? The poor quality of the aerial images used did not allow me to get a higher resolution, e.g. 10 m.

Reviewer 3 Report
- Extensive editing of the English language is needed.
- Abbreviations use is inconsistent. Needs revision.
- “Weights of Evidence” is a geostatistical method (Dongli et al 2011) and not Weights of Evidence geostatistical method.
- Terrain digital models replace by Digital Terrain Models (DTMs).
- What do you mean by filtering the points representing the terrain surface using tools available in the QCoherent software?
- “Clouds of elevation points” is not a common nomenclature for “Point Cloud”.
- The geologic description of the working area lakes to references.
- What does accurate photogrammetric DEM mean?
- Justify the use of IDW for interpolation?
- How did you identify the best results of the DTM generalisation?
- Using Low-Gaussian filter to reduce errors is not well justified in LiDAR-derived DEMs.
- What is the point clouds displayed in altitude mode? Replace, e.g. you may use Point Cloud representing height/elevation values.
- No stats were provided around how good the filtering of the point cloud of vegetation and buildings was?
- More referencing is needed regarded to some affirmations (e.g., using of WoE for determining slope susceptibility to landsliding or medical diagnostics).
- Lines 162, 163 and 164 are a copy of lines 140, 141 and 142.
- Replace inventoried by surveyed.
- What are the activity models for the landslides? Explains with mode details
- Why is the big difference between the ratio/index calculated and the ratio/index of the literature for the Dynow Foothills location?
- The chi-square results are not well presented and justified. The data division for training and testing needs justifying? And how it was done, e.g. random?
- Rasterising the streamlines and using it in the susceptibility analysis may not represent the effect of this attribute on the working procedure. A buffer of proximity to streams with weighted evaluation should approximate better to the landscape analysis regarding the susceptibility analysis.
- The selection of some topographic attributes is not well justified and more revision on topographic attributes derived from DEM is needed.
- What is the “Conditional independence test?
- Tables are not well structured. A landscape could be of good use in this case.
- Dimensionality analysis could be useful to identify important attributes (e.g. Principle component analysis (PCA)).
- Section 4.1 from results need rewriting or more development, not clear what you want to tell.
- In Figure 10, the pixel size on the maps doesn’t feel as 20X20m. Am I missing something here?
- Vegetation density and type could be a source of error in both DEMs. Don’t forget the errors generated from the interpolation process itself.
- The DEM comparison section and the differential elevation model in Results are a simple comparison of a well-known problem but don't draw any new results nor connect to the benefits of the principle work, which is "The Landslide susceptibility".
- Based on your current results, you can’t confirm that uses of LiDAR DEMs are mandatory for landslide susceptibility modelling. What about other DEM sources, e.g. photogrammetry?
- The DEM_LPIS could be greatly enhanced if you use other interpolation techniques, e.g. Kriging or correctly enforced DEMs by Hutchinson (1988-2011).
Author Response
Dear Reviewer 3
Thank you for the helpful review and many pertinent comments. You asked me a lot of questions. I have included almost all of your suggestions in the revised munuscript. Let me answer your main questions
- What do you mean by filtering the points representing the terrain surface using tools available in the QCoherent software? What I mean here is using and testing both IDW and Adaptive TIN interpolation algorithms. Ultimately, however, I chose the adaptive TIN algorithm. It was difficult as both algorithms are nearly equivalent.
- Justify the use of IDW for interpolation?
I thought about it and decided to use the Adaptive TIN algorithm. . I have described the methodology section in more detail.
- What does accurate photogrammetric DEM mean?
This means that it shows small values of the x, y, z coordinates. The vertical value is especially important. He often compares various digital terrain models to the morphology and hypsometry of topographic maps, for example in the scale of 1: 10,000. This allows me to visually verify whether the photogrammetric model is correct.
- How did you identify the best results of the DTM generalisation?
I used topographic maps for this purpose.
- Using Low-Gaussian filter to reduce errors is not well justified in LiDAR-derived DEMs.
There are different views on this, which are presented in world publications. I relied on the publication -Milledge G D., Stuart L., Warburton J. Digital filtering of generic topographic data in geomorphological research. Earth Surface Processes and Landforms, 2009, 34.1, 63 - 74
- What are the activity models for the landslides? Explains with mode details
Active slopes of landslides were located using a RTK GPS receiver. The new fractures observed both on the landslide surface and on buildings and roads were the main criteria for landslide activity. Many landslides had new fractures in the main and minor scarpes. I imported the obtained measurements into the ArcGis program and applied them to the mapped landslide surfaces.
- Why is the big difference between the ratio/index calculated and the ratio/index of the literature for the Dynow Foothills location?
Because more landslides have been inveterated than previously reported in the literature.
- The selection of some topographic attributes is not well justified and more revision on topographic attributes derived from DEM is needed.
Of course, more factors can be given. This is a good idea. However, he wants to do so in another research project. I plan to test various machine learning methods with many predictors.
- What is the “Conditional independence test?
Obviously the chi square test
- In Figure 10, the pixel size on the maps doesn’t feel as 20X20m. Am I missing something here?
It is not really a pixel with a resolution of 20m.
- Vegetation density and type could be a source of error in both DEMs. Don’t forget the errors generated from the interpolation process itself.
I agree. I wrote about this in the chapter conclusions
- Based on your current results, you can’t confirm that uses of LiDAR DEMs are mandatory for landslide susceptibility modelling. What about other DEM sources, e.g. photogrammetry?
- The DEM_LPIS could be greatly enhanced if you use other interpolation techniques, e.g. Kriging or correctly enforced DEMs by Hutchinson (1988-2011).
I agree with point 29 and 30. LiDAR data also contains a lot of errors. It depends on the scan density and the time of year of the raid. Of course, the methods of interpolation and classification also play a large role. You have to test it with better and better algorithms. Other photogrammetric data are also important. Better and more accurate aerial photos are made from which quite good digital models of the terrain can be generated.

Reviewer 4 Report
The research has been well organized and structured but the manuscript needs some improvement in order to be published.
At first the introduction is not adequately written: it should give introductory informations about the whole topic, without entering in unnecessary details regarding the study area, for instance. On the other hand it should introduce the topic from a wide view, moving to the problem and to the solution and proper references should be used.
Some lacks in references has been detected even in the methods section, as highlighted in the annotated version of the manuscript.
Finally, the Discussion section could be expanded and provide more informations and comparisons.
Further, some other points are in the annotated version.

Author Response
Reviewer 4
Thank you for your thorough review and any constructive comments.
All the suggestions you made in the manuscript have been corrected. I also rebuilt the introduction and added a lot of world literature. I also rebuilt the methodology section and added literature. In the discussion section, I also added world literature. I deleted the last sentence in the discussion section. It required a lot of development and reflection. I was afraid that I would write too much. I think he will refer to it in the next paper.

Round 2
Reviewer 1 Report
Dear Author,
Albeit considering your sound arguments in your “author response” I still have concerns regarding the general methodical approach in your manuscript (comparison of two digital elevation models, LPIS and ALS with an elevation resolution of 20 m), yet the other reviewers do not share my concerns. I do believe, however, that the overall quality of your manuscript has considerably increased with all the changes and the rework that has been applied. Therefore, my overall recommendation is to accept this manuscript in present form.
Author Response
I have attached two versions of the article. The first revision after my corrections (detailed changes are described for this revision) and the second final revised linguistic revision.
Figure.1
I changed the size of the explanation and the size of the strike and dip.
Figure 2
Has been removed.
Figure 2 (3-old numbering)
I added a map with landslide scarp activity.
Figure 6 (7-old numbering)
I added the text “Test chi-square” to text Conditional Independence.
Figure 7
I added a map with the location of the orthophotomap.
Figure. 10
I added the landslide susceptibility scales.
Figure.11
I added a new figure. Histogram representing the distribution of observed landslides falling into various susceptibility classes of di ff erent landslide susceptibility zonation maps between LSM_ALS I LSM_LPIS.
Table 1 and Table 2
Both tables have been formatted. Rows of numbers are aligned. There is only a display issue with Track Changes. The tables look bad. On the other hand, when we turn off tracking changes. Everything is ok.
- Track Changes
- Without Track Changes
Abstract.
11,14,16,19,24- I Changed terrain into elevation.
15,21, 23 I Changed modelling into modeling
13-14 I added - Inventory of landslides was performed using 1m LiDAR Dem and field research.
19 – I changed- ArcGis into ArcGIS
22- I changed geostatisticis into a geostatistical
21, 22 -deleted sentence (Track changes).
Introduction
34, 35 – I added- Landslides are a significant geodynamic threat in many areas of the world. They cause many economic losses as well as threaten human life.
35 – I added - in Poland.
36 I Changed Dynow into Dynów.
37-43 - Deleted sentences (Track changes).
45, 46,60- I Changed terrain into elevation.
46, 89- I Changed modelling into modeling.
68-87. I added - DEMs are often generated using data obtained from different remote sensors, including optical imaging sensors, light detection, and ranging (LiDAR), and synthetic aperture radar (SAR) [18]. The qualities of DEM-derived factors often depend on the spatial resolution of DEMs. This has been widely discussed in the literature [19]. Therefore, the choice of DEM is important for the assessment of landslide susceptibility.
The quality of the DEMs is essential for assessing their suitability and determines the quality of the geomorphometric analysis [20-22]. This multitude of available DEMs calls for their verification. It is necessary to remember that working with digital data requires paying particular attention to their quality. Small errors in DEMs can produce large errors in derived elevation attributes [23], especially second-order derivatives such as curvature [24]. DEM accuracy depends on the type of topography and ruggedness of the elevation as well as the type of vegetation [25], methods for collecting elevation data, method for DEM generation, type of DEM grid, and DEM resolution [26,27]. The issue of error analysis in DEMs is still current and brought up in literature [28, 29]. It was decided to focus my study on the usefulness of different DEM with this same resolution for landslide susceptibility studies. These analyses consisted of checking their vertical accuracy.
Studies on the impact of the quality of elevation digital models obtained from various source data on the result of landslide susceptibility model have not yet been widely discussed in professional literature. In my opinion, the result of landslide susceptibility modeling is more influenced on the quality of data (especially digital elevation models) than the number of factors used.
- Deleted sentences (Track changes).
- I added -The into Weights-of-Evidence) and is into geostatistical.
94-96 - Deleted sentences (Track changes).
Study Area
99 - I changed Dynow into Dynów.
100- I changed terrain into elevation.
110 I changed - Figure 1. Study area location and geological setting. (Track changes).
111 - I Changed favour into favor.
115,116,117, 118, 120 - I Changed shales- shale’s
124- I Changed Chyrzynki into Chyrzynka
124 - I Changed Dynow into Dynów.
- Methods and Materials
3.1. Archival Digital Stereo-Pair Aerial Images
141- I changed error of fitting the photogrammetric model into the grid are into error of fitting the photogrammetric model into the grid is presented.
145- I changed terrain into elevation.
150 - I changed metre into meter
151- Deleted sentences (Track changes).
- I added – point.
154 I added – cloud and ground
154 Deleted sentences (Track changes).
157-158. Deleted sentence (Track changes).
158-170- I added - The next step was the interpolation of elevation points using algorithm. Two algorithms were tested: IDW (Inverse Distance Weighting) and the Adaptive TIN model algorithm. Both algorithms provide similar classification of point clouds describing land use for agriculture, areas on which a single building, shrub or tree is located. The Adaptive TIN algorithm works better then IDW algorithm in terms of points recorded by the laser beam being reflected from ground, vehicles and bridges. The Adaptive TIN Ground Point Cloud Task is an automated algorithm designed to separate points that have a high probability of being ground points from other points. The algorithm divides the task area into cells whose X and Y dimension are defined by a Seed parameter. For each cell, a "best" candidate ground point is selected. These "seed" points are then used to construct a Triangulated Irregular Network (TIN). The algorithm then iterates, attempting to add additional points to the TIN, based on several inclusion criteria. The iterations conclude when either the user specified stopping criteria is met or no additional points are added during the previous iteration.
171 - I changed 0.5m into 1,0m
- I changed modelling into modeling.
3.3. Gaussian Low-Pass Filter
174-175 I added- Based on the literature [62] data, the Gaussian Low-Pass filter was selected for further generalization of the surface of both digital elevation models.
180- I Changed terrain into elevation.
182-184 I added -The obtained results of elevation generalisation were compared in Arc Gis software with the hypsometry of the topographic map. It was assumed that the topographic map at the scale of 1: 10,000 is a sufficiently accurate reflection of the topography.
3.4. Weights of Evidence Method
210-214- Deleted sentences (Track changes).
(3.5).4.0 Results
4.1 Landslide Inventory
228- I changed terrain into elevation.
230-248 I changed inventoried into surveyed.
230- I added sentences- In order to develop a model of landslide activity, three field trips were made. Active slopes of landslides were located using a RTK GPS receiver. The new fractures observed both on the landslide surface and on buildings and roads were the main criteria for landslide activity. Many landslides had new fractures in the main and minor scarpes. This mainly indicated the activity of landslides.
For comparison purposes, the landslide ratio/index and the landslide density index were calculated [2].
X 100%
Where,
Op - Landslide ratio/index [%]
PO - Landslide area [km2]
Pt - Surface area [km2]
Where,
G - Landslide density ratio/index [number/ km2]
n - Number of landslides [number]
Pt. - Surface area [km2]
252- I added- The reason for such a large difference was the greater number of inventory landslides.
261- I added - and active scarpe.
269- I changed stratal into strata
290- Ichanged variegated into Variegated.
(3.6) 4.2. Landslide Conditioning Factors and their Analysis
301,323 - I changed modelling into modeling.
304 - I added - and roughness
307- I added - randomly
310- 315 - I added sentences- The modeling was carried out using the ArcSDM (Spatial Data Modeler) module, which is an extension of ArcGIS, developed by the Geological Survey of Canada [72].This module automatically calculates test chi-square positive and negative weight values, variances, contrast, and posteriori probability (Table 1 and Table 2). The first three factors were created in a vector form in the geobase, while the remaining six were developed as a result of processing of digital elevation models ALS-Dem and LPIS-Dem.
316-317 - Deleted sentences (Track changes).
317-321- I added sentences - In the second stage, a chi-squared test of independence was performed for the nine predictive factors. The test chi-square is based on a comparison of the observed values (obtained in the study) and theoretical values (calculated assuming that there is no relationship between the variables). Large differences indicate the existence of dependencies between the variables.
325-328 - Deleted sentences (Track changes).
329-332- Deleted sentences (Track changes).
4.2. Comparison of the Quality of elevation differences between Digital Terrain Elevetion Models ALS and LPIS
355 – 356- I changed terrain into elevation.
361-362 - I added sentences - This research showed spatial distribution of elevation changes between Dem-ALS and Dem-LPIS.
384- Deleted -Morphological cross-section through the first area.
4.3. Landslide Susceptibility Maps
388- I changed modelling into modeling.
388- I changed terrain into elevation.
397- I changed flattenings into flattening.
400 - I changed shale into shale’s.
403-404 – Deleted sentence (Track changes).
404 – I added- number of most weakly susceptible areas occurs
413-414- I added- The modeling results showed that the greatest number of landslides occurs in the class high (12, 9%) and very high (29, 03%) on the LSM_ALS model (Figure 11).
416-417- I added - Figure 11. Histogram representing the distribution of observed landslides falling into various susceptibility classes of different landslide susceptibility zonation maps between LSM_ALS I LSM_LPIS.
- Discussion
436, 440,447,452,455 - I changed terrain into elevation
456- I changed IDW into Adaptive TIN
462 - I changed modelling into modeling.
469-471- Deleted sentence (Track changes).
- Conclusions
474, 475 - I changed modelling into modeling.
480 - I changed artefacts into artifacts.
481-482 - Deleted sentence (Track changes).
482-487- I added- The precision that can be achieved by photogrammetric measurements is mainly dependent on the quality of the aerial photographs. Photographic resolution is a function of the optical quality of an image, and influenced by the resolving power of the film and camera lens, image motion during exposure, atmospheric conditions and the conditions of film processing.
491-494- I added- It is worth noting that the density and type of vegetation may be a source of errors in both digital elevation models. On the other hand, errors in the Dem_ALS model may be generated both in the process of point cloud classification and their interpolation.
496-498- I added- The presented research helps to better understand the process of the formation of various errors and their impact on the final result of landslide susceptibility modeling.
502- I added- and machine learning.
503 - I added- different resolutions.
503- I deleted – resolution.
References
506- I added reference -
- Aleotti, P., Chowdhury, R. Landslide hazard assessment: summary review and new perspectives. Bulletin of Engineering Geology and the Environment, 1999, 58, 21 – 44.
509 – I changed - Geomorphologica into Geomorphological.
510- I changed - Polich into Polish.
514-515- Deleted reference (Track changes).
I moved from 539-541- into 532-534(Track changes).
554- 582 - I added references.
- Wang, Q.; Guo, Y.; Li, W.; He, J.; Wu, Z. Predictive modeling of landslide hazards in Wen County, northwestern China based on information value, weights-of-evidence, and certainty factor. Geomat. Nat. Hazards Risk 2019, 10, 820–835.
- Schlögel, R.; Marchesini, I.; Alvioli, M.; Reichenbach, P.; Rossi, M.; Malet, J.P. Optimizing landslide susceptibility zonation: Effects of DEM spatial resolution and slope unit delineation on logistic regression models. Geomorphology 2018, 301, 10–20.
- Desmet P.J.J., Effects of interpolation errors on the analysis of DEMs. Earth Surface Processes and Landforms,1997, 22, 563–580.
- Florinsky I.V., Accuracy of local topographic variables derived from digital elevation models. International Journal of Geographical Information Science, 1998, 12, 47-61.
- Chaplot V., Darboux F., Bourennane H., Leguédois S., Silvera N., Phachomphon K., Accuracy of interpolation techniques for the derivation of digital elevation models in relation to landform types and data density. Geomorphology, 2006, 77, 126–141. DOI: 10.1016/j.geomorph.2005.12.010.
- Fisher P.F., Tate N.J., Causes and consequences of error in digital elevation models. Progress in Physical Geography, 2006, 30, 467-489. DOI:10.1191/0309133306pp492ra.
- Wise S.M., The effect of GIS interpolation errors on the use of DEMs in geomorphology. In: Lane S.N., Richards K.S., Chandler J.H. (eds), Landform Monitoring, Modeling and Analysis. Wiley, Chichester, 1998, 139–164.
- Thomas J., Prasannakumar V., Vineetha P., Suitability of space-borne digital elevation models of different scales in topographic analysis: an example from Kerala, India. Environmental Earth Sciences, 2015, 73,3, 1245-1263. DOI: 10.1007/s12665-014-3478-0.
- Zhang W., Montgomery D., Digital elevation model grid size, landscape representation, and hydrologic simulations. Water Resources Research, 1994, 30, 4, 1019–1028.
- Li Z.L., A comparative study of the accuracy of digital terrain models (DTMs) based on various data models. ISPRS Journal of Photogrammetry and Remote Sensing, 1994, 49, 2-11.
- Bolstad P.V., Stowe T., An evaluation of DEM accuracy: Elevation, slope, and aspect. Photogrammetric Engineering and Remote Sensing, 1994, 60, 1327-1332.
- Edson C., Wing M.G., LiDAR Elevation and DEM Errors in Forested Settings. Modern Applied Science, 2015, 9,2, 139-157. DOI: 10.5539/mas.v9n2p139.
652-653 –I added reference –
- Milledge G D., Stuart L.,Warburton J. Digital filtering of generic topographic data in geomorphological research. Earth Surface Processes and Landforms, 2009, 34,1, 63 – 74.
661-669 - I added reference-
- Weed, D. L. Weight of evidence: a review of concept and methods. Risk Anal., 2005, 25, 1545–1557.
- Dongli Fan, Xi-min Cui, De-bao Yuan, Jiafeng Wang, Jinlin Yang, Shengyao Wang. Procedia Environmental Sciences , 2011, 11, 1412 – 1418.
- Dahal RK, Hasegawa S, Nonomura A, Yamanaka M, Masuda T, Nishino K. GIS-Based Weights-of-Evidence Modelling of Rainfall-Induced Landslides in Small Catchments for Landslide Susceptibility Mapping Environ Geol 54(2) (Springer) , 2008, 311–324.
- Schulz, W. H. Landslides mapped using LIDAR imagery, Seattle, Washington. US Geological Survey Open-File Report, 2004, 2004-1396, 11 p., U.S. Department of the Interior, U.S. Geological Survey, Washington D.C.
687-692 -I added reference-
- Barbieri G.; Cambuli P. The weight of evidence statistical method in landslide susceptibility mapping of the Rio Pardu Valley (Sardinia, Italy). In: Proceedings of the 18th World IMACS/MODSIM Congress Cairns Australia, 2009, 13–17.
- Bettian N.; Birgit T. Landslide susceptibility assessment using weights-of evidence applied to a study area at the Jurassic escarpment (SW-Germany) Geomorphology, 2007, 86,12–24.

Reviewer 2 Report
The author has revised a lot from original manuscript. I have no further question.
Author Response

(The authors gave the same response as above.)

Reviewer 3 Report
- Consider the English language editing
- Consider the usage of the abbreviation within the manuscript.
Author Response

(The authors gave the same response as above.)

Reviewer 4 Report
The manuscript has been improved and still some minor elements need to be adjusted, according to the attached annotated version.

Author Response

(The authors gave the same response as above.)
